# The Use of Silica Microparticles to Improve the Efficiency of Optical Hyperthermia (OH)

**DOI:** 10.3390/ijms22105091

**Published:** 2021-05-11

**Authors:** O. Casanova-Carvajal, M. Zeinoun, A. L. Urbano-Bojorge, F. Bacha, J. Solera Livi, E. Agudo, G. Vargas, M. Ramos, R. Martínez-Murillo, J. J. Serrano-Olmedo

**Affiliations:** 1Centro de Tecnología Biomédica (CTB), Campus de Montegancedo, Universidad Politécnica de Madrid (UPM), Pozuelo de Alarcón, 28223 Madrid, Spain; oscar.casanova@ctb.upm.es (O.C.-C.); michael.zeinoun@ctb.upm.es (M.Z.); lorena.urbanob@upm.es (A.L.U.-B.); fernandobacha@gmail.com (F.B.); java_jsl@hotmail.com (J.S.L.); estheragudogarcia@gmail.com (E.A.); gemadvargas@gmail.com (G.V.); milagros.ramos@ctb.upm.es (M.R.); 2CIBER de Bioingeniería, Biomateriales y Nanomedicina (CIBER-BBN), 28000 Madrid, Spain; r.martinez@cajal.csic.es; 3Escuela Técnica Superior de Ingeniería de Telecomunicación, Universidad Rey Juan Carlos, C/Tulipán s/n, 28933 Madrid, Spain; 4GDAF-UC3M, Department Tecnología Electrónica, Universidad Carlos III de Madrid, Avda. Universidad, 30. Leganés, 28922 Madrid, Spain; 5Neurovascular Research Group, Department of Translational Neuroscience, Instituto Cajal, CSIC, Av. Dr. Arce 37, 28002 Madrid, Spain

**Keywords:** optical hyperthermia, silica microparticles, cell viability, gold nanorods, optical density, cancer

## Abstract

Although optical hyperthermia could be a promising anticancer therapy, the need for high concentrations of light-absorbing metal nanoparticles and high-intensity lasers, or large exposure times, could discourage its use due to the toxicity that they could imply. In this article, we explore a possible role of silica microparticles that have high biocompatibility and that scatter light, when used in combination with conventional nanoparticles, to reduce those high concentrations of particles and/or those intense laser beams, in order to improve the biocompatibility of the overall procedure. Our underlying hypothesis is that the scattering of light caused by the microparticles would increase the optical density of the irradiated volume due to the production of multiple reflections of the incident light: the nanoparticles present in the same volume would absorb more energy from the laser than without the presence of silica particles, resulting either in higher heat production or in the need for less laser power or absorbing particles for the same required temperature rise. Testing this new optical hyperthermia procedure, based on the use of a mixture of silica and metallic particles, we have measured cell mortality in vitro experiments with murine glioma (CT-2A) and mouse osteoblastic (MC3T3-E1) cell lines. We have used gold nanorods (GNRs) that absorb light with a wavelength of 808 nm, which are conventional in optical hyperthermia, and silica microparticles spheres (hereinafter referred to as SMSs) with a diameter size to scatter the light of this wavelength. The obtained results confirm our initial hypothesis, because a high mortality rate is achieved with reduced concentrations of GNR. We found a difference in mortality between CT2A cancer cells and cells considered non-cancer MC3T3, maintaining the same conditions, which gives indications that this technique possibly improves the efficiency in the cell survival. This might be related with differences in the proliferation rate. Since the experiments were carried out in the 2D dimensions of the Petri dishes, due to sedimentation of the silica particles at the bottom, whilst light scattering is a 3D phenomenon, a large amount of the energy provided by the laser escapes outside the medium. Therefore, better results might be expected when applying this methodology in tissues, which are 3D structures, where the multiple reflections of light we believe will produce higher optical density in comparison to the conventional case of no using scattering particles. Accordingly, further studies deserve to be carried out in this line of work in order to improve the optical hyperthermia technique.

## 1. Introduction

Nanoparticle hyperthermia is a promising treatment that offers functionalities not possible using conventional magnetic materials. [1]. Hyperthermia refers to the application of heat to destroy carcinogenic cells through protein denaturation and the rupture of cellular membranes [2,3]. Currently, one of the different hyperthermia methods under investigation is magnetic hyperthermia, which even though it is not part of this research, is one of the lines in which we have experience; it is based on a localized increase in temperature, normally using excited iron nanoparticles by external magnetic fields [4,5,6,7,8,9]. The other method is optical hyperthermia (OH), which uses laser light because of its characteristics of monochromaticity, coherence, and collimation [10]. The near-infrared region (NIR) wavelength is commonly used because of its deep penetration into biological tissue with low absorption by water molecules and hemoglobin in this spectral region [11].

Gold nanoparticles are used in OH due to their resistance to corrosion, low toxicity, biocompatibility, and ease to manage for conjugating biocompatible ligands to their surface. Optical characteristics of gold nanoparticles are as well helpful for OH because they are highly photostable, exhibit an enhanced absorption cross-section, offer high light-to-heat conversion, and possess excellent optical absorption variability, which can be tuned over a wide range of wavelengths [12]. Following light absorption, the resulting heat is dissipated within picoseconds toward the surrounding medium [13], giving rise to photo thermal damage to biological tissues, including cell membrane ruptures, protein denaturation, and the impairment of DNA and RNA synthesis [2,3]. A CEM43 °C thermal dose threshold is used for the measurement of hyperthermia with tumoricidal effects. A study performed on dogs demonstrated the use of the CEM43 °C scale is proportional to the time in minutes, where 2 CEM43 °C scales are equivalent to 2 scale minutes. To reduce CEM43 °C’s side effects, optical hyperthermia should be optimized to safe doses [14] and the concentration of gold nanoparticles reduced to decline their toxicity [15,16], while the heating efficiency for the therapy should remain unaltered. There are various types of gold nanoparticles, including gold nanospheres, nanoshells, nanostars, and others. In this study, GNRs were used because they exhibit the highest absorption in their cross-section among all gold nanoformulations; two types of wavelength are distinguished, longitudinal and transversal [17], with a ratio variation that tunes their absorption within the NIR; and for their small sizes 20 nm and 40 nm [12,13,18].

The possibility to introduce a complementary agent for improvement of the GNRs properties in the OH was explored, taking into consideration the ability of the agent to reduce both the GNRs concentrations in the medium and the laser radiation dose but maintaining the effect of the treatment. This complementary agent should have a shape and size with capacity to scatter light that travels through the tissue that holds the GNRs; it is capable of increasing light interactions with GNRs so as to magnify the optical density of the tissue, thus experiencing a rise in temperature when irradiated. We selected SMSs as they possess light-scattering characteristics, are highly dispersible, and are biocompatible with lightweight materials [19,20]. At certain volumes, SMSs could reduce the need for GNRs, allowing us to reduce their concentration and toxicity, regulate optical density in a local way, and maintain the therapy’s temperature. The experimental design seeks to establish the relationship between the scattering agents and the absorbent particles to modulate heating and thus be able to optimize cell mortality rates of cell lines in an in vitro experiment.

Silica is generally recognized as safe (Food and Drug Administration, FDA). Silica, in the form of Cornell dots (C dots), received FDA approval for stage 1 human clinical trials for targeted molecular imaging [11,21]. Silica particles have been described by He et al. as drug carriers for poorly soluble drugs and gene delivery, for both viral and nonviral systems [22]. Zhang et al. also considered silica’s biocompatibility characteristics [23]. In this study, SMSs were used to determine variations in the toxicity rate after conducting in vitro cytotoxicity tests at 24 h [24]. Likewise, experiments were carried out to prove silica’s biocompatibility when mixing with biological cells, such as the CT-2A and the MC3T3-E1 cell lines established from C57BL/6 mice [20,25].

In the search for a promising biocompatible agent to induce light transformation in heat for cancer therapy, the objective of the study presented here is to show our first results in the research around the contribution of SMSs to OH for cancer with GNRs. In particular, here, we aim to test the ability of silica to compensate reductions in GNRs concentration, thus favoring lesser toxicity and lower laser beam power but maintaining photothermal damage. To reach this objective, SMSs were placed together with the GNRs in the cell culture medium for (i) glioblastoma CT-2A and (ii) osteoblastic MC3T3-E1 cells. After cell incubation in appropriate cell culture media, the preparation was subjected to laser radiation treatment. The cell survival was analyzed to determine the efficacy of the OH. Whether silica could be a promising biocompatible agent to induce light transformation in heat for cancer therapy is the main subject of this research.

## 2. Results

### 2.1. In Vitro Cell Viability Experiments

We previously reported that tumor cells are consistently more sensitive to heat than healthy cells [25]. Thus, the effect of optical hyperthermia on CT-2A tumor and non-tumor MC3T3-E1 cells was studied by examining differences in the heat resistance between them. In addition, we analyzed the effectiveness of the radiation fostering cell death when cell culture media is supplemented with gold nanoparticles and/or SMSs. The results obtained from these experiments, regarding both cell lines, are presented separately.

#### 2.1.1. CT-2A

Figure 1 and Figure 2 show the results of a number of experiments, which were divided into four groups according to the four different experimental conditions: Control, SMSs, GNRs, and SMSs + GNRs. Following the laser radiation of all groups, the results obtained were photographed following the viability staining protocol reported above. Next to the pictures, we have placed a bar diagram to clearly show the percentage of cell death in each of the four groups and the low standard deviation. The first column in each bar diagram (blue) in Figure 1 and Figure 2 represents the Control group cell damage caused by laser beam radiation. The orange and gray columns represent cell damage count in cell culture treated with SMSs and GNRs, respectively. Finally, the yellow column represents cell death number in the SMSs + GNRs group, where the concentration of GNRs was reduced to 80% with respect to the GNRs column, and the number of SMSs microparticles was increased to 20%. Notice that a significant reduction of the GNRs in the incubation medium solution did not alter significantly the killing cell capacity of the experimental therapy.

The radiation period was kept constant at 10 min in all the experiments. To observe the constant time parameters at different power densities, two different radiation power densities were applied, 1000 mW/cm^2^ (Figure 1) and 900 mW/cm^2^ (Figure 2). Note that applying higher power densities, the cell damage was greater, as shown in the graph by 8.27% and 5.19% cell mortality at 1000 and 900 mW/cm^2^, respectively. The SMSs solution group had a slightly higher mortality percentage under the same conditions, revealing that SMSs is useless for OH when used alone. This result runs contrary in the case of GNRs, where the cell mortality percentage increased to 86.04% and 66.57% at 1000 and 900 mW/cm^2^, respectively. It is noteworthy that upon decreasing GNR concentration by 20% and introducing SMSs (SMSs + GNRs group), the value of cell mortality increased to 95.46% and 89.023% at 1000 and 900 mW/cm^2^, respectively. The latter shows that the dispersing agent silica not only compensated but even enhanced the heating efficiency of the GNRs.

The data obtained for cell death rate between 1000 and 900 mW/cm^2^ radiation power over a 10 min period time showed no huge group differences, except for the GNRs group. A 20% difference was observed only in the pure GNR group. In the SMSs + GNR group, the difference between high and low power density was not considered vital, as the lowest mortality percentage was around 90%.

#### 2.1.2. MC3T3-E1

As in CT-2A experiments, studies with the MC3T3-E1 cell line were divided into four groups: Control, SMSs, GNR, and SMSs + GNRs. Two different radiation power densities, 1000 mW/cm^2^ (Figure 3) and 900 mW/cm^2^ (Figure 4) were used, with the radiation period constant at 10 min. The postexposure study of cell death and survival was carried out, as in CT-2A experiments. Data of the cell mortality are shown in Figure 4 and Figure 5.

Bars in Figure 3 and Figure 4 illustrate the radiation at 1000 and 900 mW/cm^2^, respectively. The Control group exhibits the lowest cell mortality percentage, with healthy cells retaining higher resistance [2,3,14]. Comparing the effect of different radiation intensities (1000 and 900 mW/cm^2^), it is clear that when switching from higher to lower power density values that in the case of the SMSs and GNRs groups, the mortality dropped from 22.01% to 4.03% and from 17.94% to 8.18%, respectively. In the case of the SMSs + GNRs group, the percentage dropped from 55.4% to 39.55%.

Taking together, the results on in vitro cell viability experiments show that exposure at 900 mW/cm^2^ is safer for healthy cells [26] than exposure at 1000 mW/cm^2^; also, the tumor cell damage (CT-2A) was clearly higher in the same conditions than MC3T3-E1 non-tumor osteoblastic cells.

### 2.2. Temperature Data

The thermal characterization of the different experimental groups was analyzed on inert samples. Studies were performed in samples lacking cells but containing GNRs and SMSs. The effect on thermal increase of radiation powers at 1000 mW/cm^2^ (Figure 5A) and 900 mW/cm^2^ (Figure 5B) are shown. Figure 5A,B show that hyperthermia temperatures of 44 °C were reached with either GNRs or SMSs + GNRs. Notice that temperature values were higher under a radiation power of 1000 mW/cm^2^.

### 2.3. Cell Viability Experiments at Various Times and Radiation Powers

The effect of laser exposure on cell viability was studied with laser exposure, at power levels 900 and 1000 mW/cm^2^, and different time exposures, as a control to evidence the optical properties of GNRs and SMSs. The results obtained from both cell lines are shown in Figure 5C,D.

#### 2.3.1. CT-2A

In Figure 5C, the red bars show the percentage of live cells placed for the CT-2A at 30, 60, 90, and 120 min without being irradiated to provide a reference sample in comparison with other radiation powers. The green bars show the percentage of live cells following exposure at 1000 mW/cm^2^ for the same exposure periods. The blue bars show the percentage of live cells following exposure at 900 mW/cm^2^ for similar exposure times. These in vitro results indicate that longer exposure time and higher power laser exposure result in lower cellular viability.

#### 2.3.2. MC3T3-E1

Figure 5D shows the results of the cell viability for the MC3T3-E1 at 30, 60, 90, and 120 min after plating. Red bars show the percentage of live cells in the Control group for the different times of the experiment. The blue bars show cellular viability following a radiation power of 900 mW/cm^2^ for the different exposure times, and the blue ones indicate the cell viability following exposure at 1000 mW/cm^2^ for the different exposure times.

As observed in the CT-2A experiments, higher cell viability requires shorter exposure time and lower-power exposure, which is in contrast with low cell viability that requires longer exposure time and higher-power exposure. Notice that MC3T3-E1 cells were more resistant to exposure that CT-2A cells, and also that cancer cells are more sensitive to heat than healthy cells [25].

### 2.4. Cytotoxicity Experiments

CT-2A and MC3T3-E1 cells demonstrated similar characteristics, showing no significant differences in cytotoxicity (*p* < 0.05). To study cell survival and the mortality of cultured cells following treatment, a viability staining protocol was carried out as shown before. Neither the GNRs nor SMSs produced relevant cellular toxicity either individually or mixed. Figure 6 shows cell mortality rate at 24 and 48 h after reagents (SMSs or GNRs) were supplemented to the cell culture medium. Therefore, we can conclude that SMSs, GNRs, and the mix of both materials appears to be biocompatible, as it was proved to be nontoxic. This result correlated well with previous studies that SMSs and GNRs have been used as bioactive materials for bone prostheses in clinic [27].

## 3. Discussion

Recent studies addressed the toxicity of GNRs as limiting factors for their use in biomedical applications, including cancer treatment. Here, we provide experimental in vitro recommendations to reduce GNRs toxicity for future research in the field of biological systems [28]. In order to reduce the toxic effect of GNRs in OH, we tested the possibility to introduce in the reaction medium a complementary agent for improvement of the OH properties of GNRs. At the same time, our goal was to reduce the laser radiation dose, keeping the efficiency of the treatment by maintaining the therapy temperature. As reported in the Introduction section, we select an agent with shape and size capacity to scatter light that travels through the tissue that holds the GNRs, which is capable of increasing light interactions with GNRs so as to magnify the optical density of the tissue and thus experience a rise in temperature when irradiated. We selected SMSs as possessing these properties [19,20]. The experimental layout seeks to establish the relationship between the scattering agents and the absorbent particles to modulate heating and thus optimize cell mortality rates of cell lines in an in vitro experiment [26,29].

Biocompatibility studies were performed in vitro by using assays similar to those used in drug development screening. Viability assays assess the overall dose-dependent toxicity of nanoparticles on cultured cells, looking for cell survival and proliferation after nanoparticle exposure. The results of this study show the efficacy of SMSs promoting the tumoricidal effect of the optical GNRs hyperthermia in two in vitro cell models, CT-2A and MC3T3-E1. In this regard, it should be noted that the tumor cell line CT-2A showed higher sensitivity than MC3T3-E1 non-tumor osteoblastic cells in the same experimental conditions. The latter support the idea that non-tumor tissues near cancer cells might remain healthy or lesser affected by the treatment than proliferative cancer cells. This procedure combining SMSs and GNRs at low concentration constitutes a promising alternative choice to treat cancer, supporting future research in this field.

There are many assays used to measure the cellular impact of a drug that can also be applied to measure the impact of nanoparticle exposure on living cells. One common assay is the use of Calcein/Propidium Iodide assay, which is followed in this research by fluorescence microscopy. Although gold nanoparticles absorb light in the visible region, we found no interference with these assays.

To identify the contribution of the dispersing agent silica to optical hyperthermia, different concentration ratios of silica and of radiation power density values were used alone or in combination with GNRs. Regarding our results on cell viability at various exposure times and radiation powers, we found a reliable harmful effect related to long-term laser exposure at 1000 mW/cm^2^ power density for both cell lines. The results obtained show a direct relationship between power density and cell damage, 8.27% and 5.19% cell mortality for CT-2A cells after 10 min exposure at 1000 and 900 mW/cm^2^, respectively. The pure silica preparation group had a slightly higher mortality percentage under the same conditions, 10.35% and 6.67% after 10 min exposure at 1000 and 900 mW/cm^2^, respectively. Therefore, we can conclude that the simple exposure of cultured cells to either laser radiation or both laser radiation and silica preparations are useless for OH. In the same experiment, treatment of CT-2A cultured cells with GNRs decreased dramatically cell viability, raising the percentage of photothermal destruction to 86.04 and 66.57 at 1000 mW/cm^2^ and 900 mW/cm^2^, respectively. By combining GNRs and SMSs in the same preparation, we found a rough decrease of cell viability; characteristically, after 10 min of exposure, the photothermal destruction reached 95.46% and 89.023% at 1000 and 900 mW/cm^2^, respectively. It is noteworthy that in this paradigm, 900mW/cm^2^ laser power increased the percentage of death cells in the SMSs + GNRs group with a higher strength when comparing the same treatment with a laser power of 1000 mW/cm^2^: 86.04% to 95.46% at 1000 mW/cm^2^, whereas it increases from 66.57% to 89.023% at 900 mW/cm^2^. The temperature readings from samples of inert GNRs, under a 10 min exposure time with an optical density of 1000 and 900 mW/cm^2^, were strikingly lower than those obtained in mixed samples containing GNRs at the same concentration plus SMSs, suggesting that the addition of silica constitutes a practical procedure that helps to offset temperature decrease when GNRs concentration is reduced. In accordance with the experimental results, it can be concluded that the optical characteristics of SMSs allowed a decrease of GNRs concentration by 20% (0.04 mg/mL) without affecting the cell viability rate. The GNRs concentration reduction in our experimental condition yields to a reduction in toxicity by GNRs without affecting the effectiveness of the hyperthermia therapy. The period of laser radiation, which is harmful over long-term exposure, was decreased from 30 to 10 min. Similarly, the beam power density was reduced from 1000 to 900 mW/cm^2^, which is an interval showing the scarce negative influence of laser radiation on biological tissues, but maintaining a level of 90% cancer cell mortality rate. The results obtained led us to conclude that SMSs might be used to reduce the concentration of both the GNRs and power laser density to obtain photothermal efficacy for cancer OH reduction.

The effect of laser radiation on biological tissues varied in the following three variables: GNRs concentration, dose, and radiation power. A decrease in GNRs concentration is essential for lowering toxicity.

## 4. Material and Methods

### 4.1. Silica

The objective of this research is to develop a safer procedure for hyperthermia therapy based on lowering GNRs concentration and thus its toxic effects on the tissue [15,16] without affecting the effectiveness of the hyperthermic procedure against malignant cells. For this, SMSs of appropriate diameter in order to scatter the impinging photons are required. Scattered photons will follow paths inside the irradiated volume that will be larger than without scattering before they either escape from the volume or are absorbed. Since a larger path implies a larger probability to interact with the light-absorbing nanoparticles—that is, a higher optical density of the irradiated volume—a higher absorption of energy will take place and consequently, a larger temperature rise for the same dose of irradiated energy.

We have used an online calculator of scattering efficiency of spherical particles in water suspension (Mie Scattering Calculator, OMLC, Oregon Health and Science University, Portland State University, and the Oregon Institute of Technology-Oregon USA) [30] to obtain this parameter as a function of the particle diameter (Figure 7B) when using spherical silica microparticles in water suspension. The scattering efficiency *Q_S_* relates the scattering cross-section area of a given particle in a given medium, which are both mainly characterized by their respective refraction indexes, to the particle size (see Figure 7A) as follows:(1)$$\sigma_{s}=Q_{s}\;A$$
where $\sigma_{s}\;$is the effective scattering cross-section area of each scattering particle and A is the physical cross-section size of the particle. Depending on the nature of the particle, medium, and wavelength of the light, the particle can reflect more or less light so that the scattering efficiency can be greater or smaller than one as shown in Figure 7A for pure silica particles within a small range of diameters when being irradiated with light of 808 nm.

The calculation shown in Figure 7A indicates that the better particle size in our case is a diameter about 2.5 µm. The nearest particles we found used in our experiments were S particles of 3 μm in diameter with plain surfaces from KISKER BIOTECH GmbH &Co (S type at a 50 mg/mL concentration as purchased). From the provider information, the SMSs were a monodisperse suspension. Their density is of 2.0 g/cm^3^ and stable in water and organic solvents with a refractive index of 1.465 ± 0.02 nm.

### 4.2. Gold Nanorods

The GNRs used were commercial (10 × 40 nm C12-10-808-TPG-50, NANoPARTz, OD/mL: 50). According to data provided by the manufacturer, the optical density of the original stock was 50, with a concentration of 1.75 mg/mL. The optical density was decreased by dilution with 49 mL cell culture medium per 1 mL of GNR suspension to reach 1 (our standard suspension). Then, the final concentration was of 35.7 μg/mL.

### 4.3. Laser Exposure Characteristics and Temperature Measurement

#### 4.3.1. Laser Exposure

The continuous laser wave (MDL H808, PSU-H-LED power source; Changchung New Industries, Changchun Jilin, China) was used at 808 nm, with a maximum output power of 5 W, a beam height from the base of 29 mm, a beam diameter with an aperture of 5–8 mm^3^, and laser head dimensions of 155 × 77 × 60 mm^3^. The laser was connected to the system via a multimode optical fiber with a core diameter of 600 µm, a length of 1.5 m, and a power transmission of 90–99% (600 µm MM fiber; Changchung New Industries).

The optical fiber was suspended vertically through a collimation lens (78382, Newport, Irvine, CA, USA), downward facing, with tests carried out using a laser power of 1000 mW/cm^2^ (based on previous experience [25]) and 900 mW/cm^2^ at different times (10, 20, and 30 min), measured using a Newport power meter model 843-R with a Newport 818-SL photodetector.

#### 4.3.2. Temperature Measurement

For temperature measurement, a two-channel manual precision thermometer (F100, Automatic Systems Laboratories, Redhill, UK) was used and connected to a Pt100 measuring probe, a platinum-made resistive temperature detector (RTD), and sensors based on the variation of the resistance of a conductor. When metal is heated, its thermal agitation is greater, and dispersing electrons reduce their average speed, thereby increasing resistance. The platinum probe offers a stable and accurate measurement. As a thermosensitive element, this probe has a high melting point (1.772 °C), is resistant to oxidation at high temperatures, and is chemically inert at high temperatures [31].

The temperature probe was introduced into the samples without touching the bottom of the dishes; this experiment was repeated in all the temperature measurement tests. With the help of the LabView programming environment (National Instruments, Austin, TX, USA), it was developed in order to acquire temperature data, and it delivers a file compatible with data processing software; in this case, the data were processed in OriginLab^®^.

### 4.4. Cell Culture Procedure

CT-2A astrocytoma and MC3T3-E1 osteoblastic mouse-derived cell lines were obtained as a generous gift from Prof. T.N. Seyfried (Boston, MA, USA) and Sigma (MC3T3-E1 Cell Line from mouse 99072810 Sigma-Aldrich), respectively. The tumor cells were seeded in 24 flat-bottom dishes cell culture plates. CT-2A cells were diluted in RPMI 1640 medium (Gibco, Invitrogen Co., Carlsbad, CA, USA). MC3T3-E1 cells were plated in DMEM high-glucose medium. The two cell lines were supplemented with 10% heat-inactivated fetal bovine serum (FBS) in a humidified atmosphere containing 95% air and 5% CO_2_ at 37 °C. Cells grew in vitro attached to the plastic of culture well as a monolayer, which is a characteristic that is essential for their proliferation, which stops once they became confluent. Adherent cells exhibited an elongated shape. When reaching confluence, cells were trypsinized and collected. The pellet was resuspended again in the same culture medium and plated again or kept in a frozen media for further experiments.

### 4.5. Cell Experimental Treatments

Briefly, cells were plated as in Section 4.4 and allowed to grow for 24 h in a CO_2_ incubator with a Cell Density per well plate of 5000. Then, cells in culture dishes were subjected to the presence of SMSs, GNRs, or both SMSs + GNRs for 1 h. Afterwards, cells were submitted to 10 min of exposure time outside the CO_2_ incubator, which was then followed by a cell viability assay. Groups were:

Control: After 24 h incubation, cells were irradiated with laser outside the CO_2_ incubator for 10 min. This allowed us to discriminate cell deleterious effect due to laser radiation, exclusively.

Silica (SiO_2_): After 24 h incubation, SiO_2_ was added to the incubation media at a concentration of 2.5 mgr/mL. Then, the culture plates were placed in the CO_2_ incubator for 1 h followed by 10 min laser radiation phase outside the CO_2_ incubator.

Gold nanoparticles: After 24 h incubation, culture cells were placed in the CO_2_ incubator for 1 h in the presence of GNRs (28 µgr/mL), which was followed by 10 min laser radiation outside the incubator.

Silica and GNRs: This time, SMSs and GNRs solutions were used in a proportion of 4.5 mgr/mL each. After 24 h of incubation time, growing cells were placed in the CO_2_ incubator in cell culture medium with SMSs + GNRs, for 1 h. Then, plates were irradiated for 10 min outside the incubator.

#### 4.5.1. Radiation Phase

The radiation period was constant at 10 min, and the power density of the light beam was varied for optimization purposes (800–1000 mW/cm^2^).

#### 4.5.2. Postradiation Phase

After exposure, all cell culture plates were placed in the CO_2_ incubator for 24 h. Checking cell viability, cells were mixed and processed in a solution made of propidium iodide and calcein as described in Section 4.6.1. After staining, dead red cells and live green ones taken from random areas of the well surface were photographed. The images taken from each well were analyzed by ImageJ software (Image Processing and Analysis Java^®^) to estimate the percentages of cell mortality/viability.

### 4.6. Cytotoxicity

To determine cytotoxicity, cells were plated at a density of 25000 MC3T3-E1 and CT-2A cells per well with 500 µL of incubation DMEM medium. After 24 h, when reaching confluence, adherent cells were treated with culture media supplemented with reagents (GNRs or SMSs). Cell viability was tested by following the four conditions: control, or with GNRs, or with SMSs, or with GNRs + SMSs. All culture plates were placed in a CO_2_ incubator for 24 h after treatment.

#### Cell Viability

To study cell viability, a calcein-propidium iodide assay procedure was carried out. Calcein and propidium iodide to assess cellular viability or cell death, respectively, were used at the same time. For this, a solution was prepared by adding 0.5 μL calcein, AM C3100MP (Life Technologies Europe BV), and 0.5 μL propidium iodide (Sigma Aldrich) to 1 mL of cell Dulbecco’s phosphate-buffered saline (DPBS). After removing the cell culture media, 500 μL (for a 24-well plate) calcein and propidium iodide solution was added into the culture plate and incubated for 30 min at 37 °C. After incubation and gentle washing with DPBS, images from each well were captured by using a fluorescence microscope (Leica DMI3000 B) using the appropriate filters: for calcein, 485 nm (viable cells, green cells), and for propidium iodide, 530 nm (non-viable cells, red). Pictures were taken from each experiment at different locations of the dish. Then, micrographs were processed by using the free ImageJ software for cell counting of the number of cells to obtain statistical data. Dishes containing cells and treated with 0.2% Triton X-100 were used as controls to determine the background signal from death cells for both calcein and propidium iodide assays. The operator was blinded with regard to the laboratory acquiring data.

### 4.7. Statistical Analysis

Social Sciences (SPSS) software was used for statistical analysis. Where applicable, the results are displayed as the mean ± SEM (standard of error the mean) of six experiments performed independently. Statistical significance was calculated using ANOVA followed by Tukey’s post hoc test. ANOVA reports a *p* < 0.05, indicating there are significant differences in the means in the groups.

## 5. Conclusions

SMSs constitute an important tool for transforming light to heat in optical hyperthermia. Here, we demonstrated its scattering properties, biocompatibility, and efficiency in in vitro models contributing to enhancing OH of tumor cells by GNRs. The results of this research demonstrated that the contribution of SMSs to OH involves (i) an increase of the potential therapeutic efficiency of GNRs, which allows a 20% reduction in the GNRs concentration, and (ii) a dropping of the radiation time and power density by 66% and 10%, respectively. Our main conclusion is that the use of functionalized SMSs should be considered part of a newer approach for glioblastoma tumor treatment based on OH. The most important results of this research are the findings according to the data where the concentrations of GNRs are reduced and a difference in mortality was observed between CT2A cancer cells and cells considered non-cancer MC3T3, maintaining the same conditions, which gives indications that it possibly improved the efficiency in the cell apoptosis process. Nevertheless, since the experiments have been carried out using near-2D structures (Petri dishes), due to sedimentation of the silica particles at the bottom, much of the light could escape easily through the upper and downer, the medium–air and medium–plastic surfaces, without suffering further reflections, as could be hypothesized would take place in normal living tissues that have a real 3D structure. Although of course, when dealing with real living tissues, the environment is completely different from simple cell cultures, and more things must be taken into account, an overall better behavior of the technique might be possible.

## Figures and Tables

**Figure 1 ijms-22-05091-f001:**
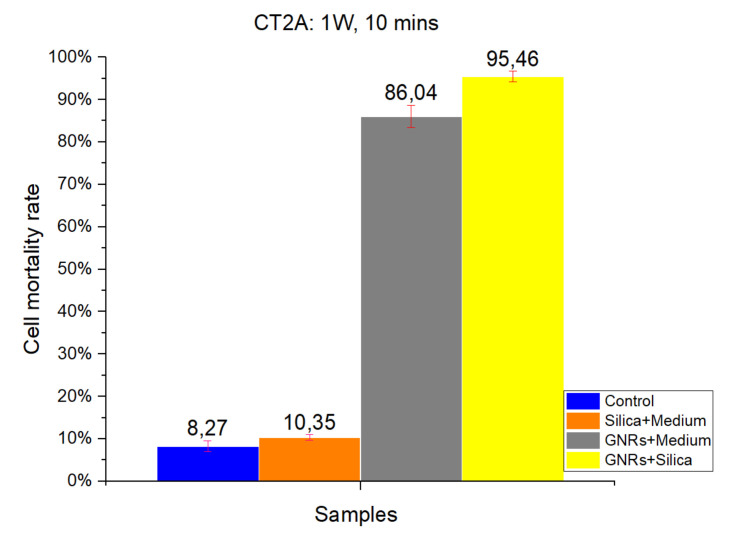
The results obtained in CT2A in vitro cell viability experiments. Radiation at 1000 mW/cm^2^ for a period of 10 min. Control Group, Blue Bar. SMSs Group (2.5 mgr/mL), Orange Bar. GNR Group (28 µgr/mL), Gray Bar. Group with GNRs (28 µgr/mL) + SMSs (2.5 mgr/mL), Yellow Bar. Data represent the mean ± SEM (standard of error the mean) of six independent experiments. ANOVA followed by Tukey’s post hoc test revealed the absence of significant differences (*p* < 0.05) in each group.

**Figure 2 ijms-22-05091-f002:**
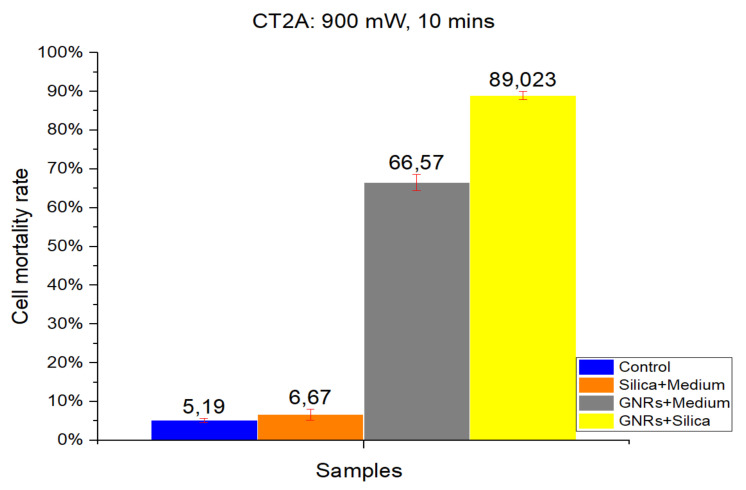
The results obtained in CT2A in vitro cell viability experiments. Radiation at 900 mW/cm^2^ for a period of 10 min. Control Group, Blue Bar. SMSs Group (2.5 mgr/mL), Orange Bar. GNR Group (28 µgr/mL), Gray Bar. Group with GNRs (28 µgr/mL) + SMSs(2.5 mgr/mL), Yellow Bar. Data represent the mean ± SEM (standard of error the mean) of six independent experiments. ANOVA followed by Tukey’s post hoc test revealed the absence of significant differences (*p* < 0.05).

**Figure 3 ijms-22-05091-f003:**
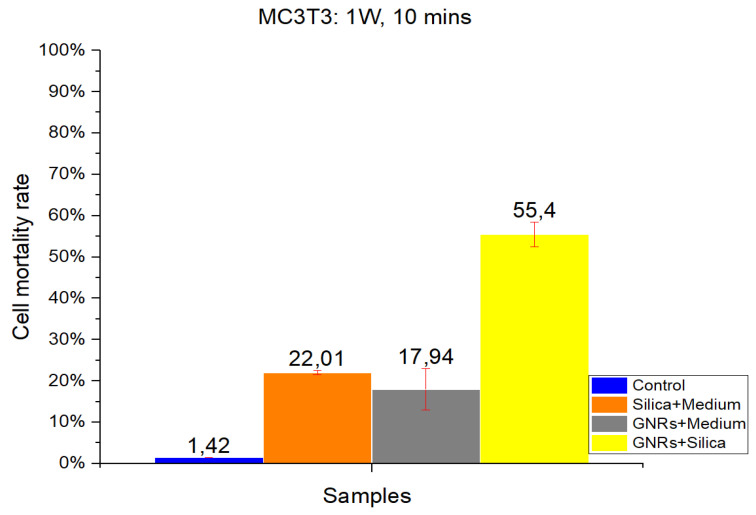
The results obtained in MC3T3 in vitro cell viability experiments. Radiation at 1000 mW/cm^2^ for a period of 10 min. Control Group, Blue Bar. SMSs Group (2.5 mgr/mL), Orange Bar. GNR Group (28 µgr/mL), Gray Bar. Group with GNRs (28 µgr/mL) + SMSs(2.5 mgr/mL), Yellow Bar. Data represent the mean ± SEM (standard of error the mean) of six independent experiments. ANOVA followed by Tukey’s post hoc test revealed the absence of significant differences (*p* < 0.05) in each group.

**Figure 4 ijms-22-05091-f004:**
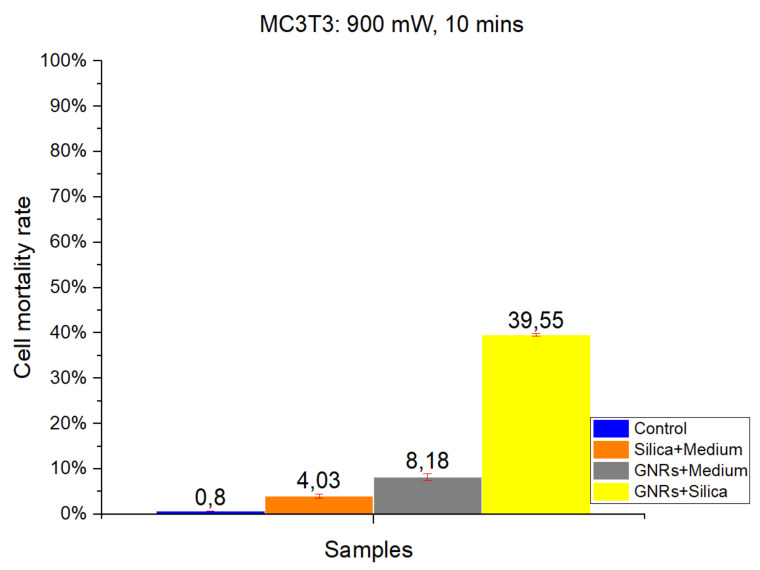
The results obtained in MC3T3 in vitro cell viability experiments. Radiation at 900 mW/cm^2^ for a period of 10 min. Control Group, Blue Bar. SMSs Group (2.5 mgr/mL), Orange Bar. GNR Group (28 µgr/mL), Gray Bar. Group with GNRs (28 µgr/mL) + SMSs(2.5 mgr/mL), Yellow Bar. Data represent the mean ± SEM (standard of error the mean) of six independent experiments. ANOVA followed by Tukey’s post hoc test revealed the absence of significant differences (*p* < 0.05) in each group.

**Figure 5 ijms-22-05091-f005:**
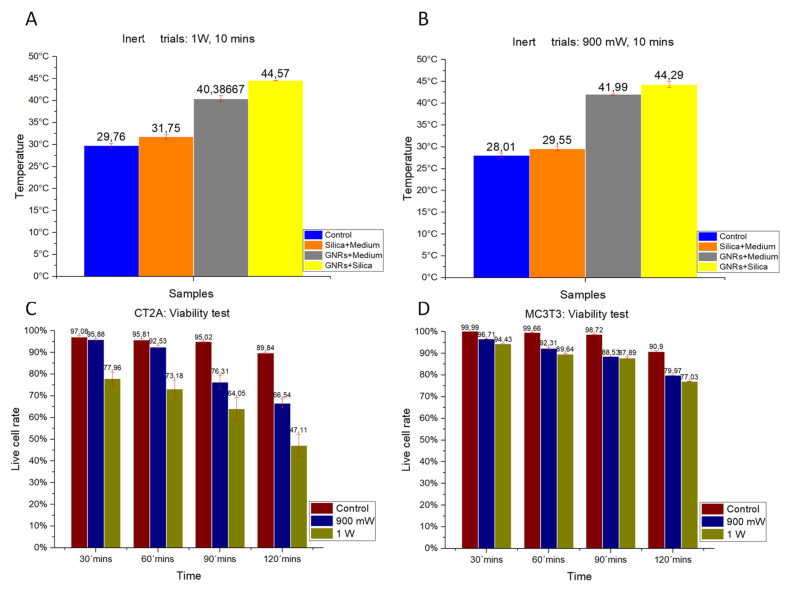
Trials using inert samples at a radiation power of (**A**) 1000 mW/cm^2^ and (**B**) 900 mW/cm^2^ for a radiation period of 10 min. (**C**). CT-2A cell viability test at different exposure times (in minutes: mins) with laser exposure. (**D**). MC3TE cell viability test at different exposure times (in minutes: mins) with laser exposure. In (**A**,**B**): Control Group, Blue Bar. SMSs Group, Orange Bar. GNR Group, Gray Bar. Group with GNRs + SMSs, Yellow Bar; in (**C**,**D**): Control Group, Red Bar. 900 mW/cm^2^ Group, Blue Bar. 1000 mW/cm^2^ Group, Green Bar. The data represent the mean ± SEM (standard of error the mean) of six independent experiments. ANOVA followed by Tukey’s post hoc test revealed the absence of significant differences (*p* < 0.05) for each group.

**Figure 6 ijms-22-05091-f006:**
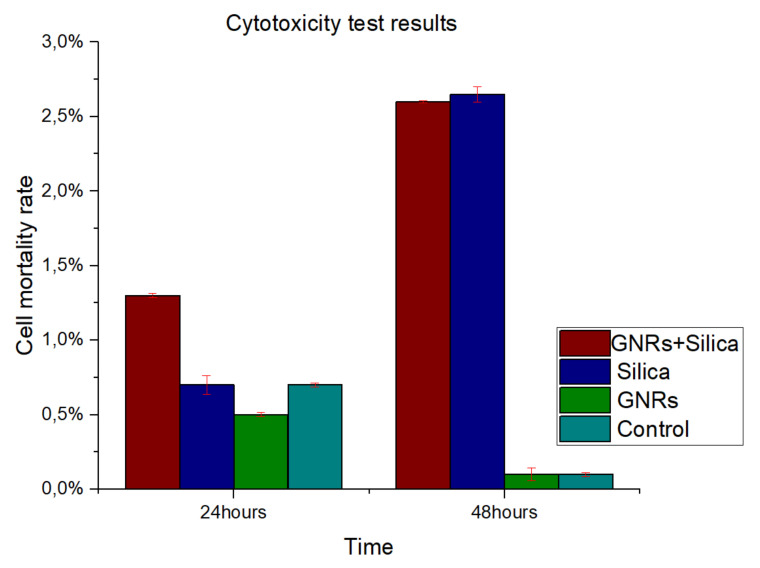
Percentage of cell mortality in observed in cytotoxicity test results with GNRs and SMSs, in addition to a range sample for a period of 24 and 48 h. Control Group, Aquamarine Bar. SMSs Group, Blue Bar. GNR Group, Green Bar. Group with GNRs + SMSs, Red Bar. Data represent the mean ± SEM (standard of error the mean) of six independent experiments. ANOVA followed by Tukey’s post hoc test revealed the absence of significant differences (*p* < 0.05) for each group.

**Figure 7 ijms-22-05091-f007:**
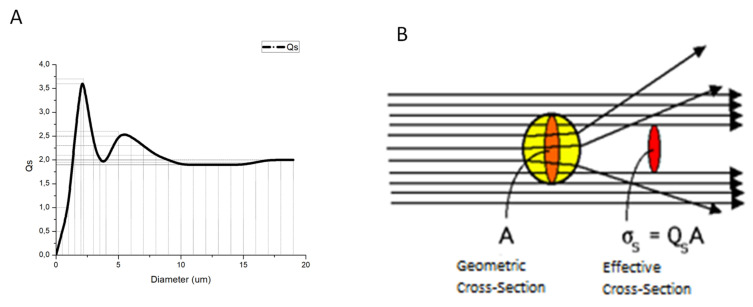
(**A**). Scattering efficiency compared to the diameter for an 808 nm continuous laser beam. (**B**). Effective cross-section concept.

## Data Availability

Not applicable.

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
