# Peer review of "The Use of Silica Microparticles to Improve the Efficiency of Optical Hyperthermia (OH)"

_ijms, 2021, doi:10.3390/ijms22105091_

Round 1
Reviewer 1 Report
The authors reported that the use of silica microparticles improves the efficiency of gold nanorods to optical hyperthermia against cancer cells. I would like the authors to address the following remarks:
Q1 - Lines 33-35: “We found a difference in mortality between CT2A cancer cells and cells considered non-cancer MC3T3, maintaining the same conditions, which gives indications that this technique possibly improves the efficiency in the cell apoptosis process”. Without additional studies evaluating the mechanisms of cell death, on what basis do the authors support this statement in the Abstract?
Q2 - The rewriting of some phrases should be considered. Full copies of the sentences from the cited works must be avoided. Example: Lines 48-51, “Hyperthermia is a therapeutic procedure used to raise the temperature of a region of the body affected by cancer. It is administered together with other cancer treatment modalities (multimodal oncological strategies). The temperature increase required can be achieved by various methods of Hyperthermia [1].
Q3 - Several problems detected in the citations. I did not find any correlation between the cited works with the statements. Examples: Lines 43-47, Refs 4-6; Lines 88-90, Line 96-98, Ref 13-14 (Suggested citation Science Translational Medicine, 2014, 6, 260ra149). Absence of references in some sentences. Examples: line 166 “based on previous experience”; line 391 “this result correlated well with previous studies...”
Q4 - For a better comparison of the results, I would suggest the introduction of a table at the beginning of that section showing the values obtained under the various conditions.
Q5 - Additional data as the concentrations should be added to the figure’s captions. The cell line was not identified in Figure 7.
Q6 - Lines 272-276: In the viability studies to assess the cumulative effect of SMSs and GNRs (SMSs + GNRs), the authors reduced the concentration of GNRs and increased the concentration of SMSs compared to the respective groups alone. Why the need to increase the concentration of SMS, therefore two variables were changed simultaneously? What is the impact of this increase on cell death?
Q7 - Please clarify the results observed in the cytotoxic evaluation for the control and GNRs groups at 24 and 48 h (Figure 7)?
Author Response
We have generated a document for the reviewer.

Reviewer 2 Report
The submitted manuscript is very interesting for hyperthermia-based cancer therapy. The discussion part is quite a significant part of this manuscript. However, there is little information needed for clarification. For example-
- Optical absorbance properties like UV-visible-NIR spectroscopy are required for 808 nm laser responsiveness. The selection critter of 808 nm laser should be mention in the manuscript.
- Cells density per well plate should mention in the experimental section.
- The concentration of GNR and silica nanoparticles should mention in the caption of each figure.
- In the 2.6.1 section, the authors have mentioned that they use fluorescence microscope for cell viability study. As the facility is available, the authors could consider adding in vitro bioimaging results to rectify cell viability more accurately.
- Properly organize the results would help the reader correlate changes during similar treatments. The authors should organize their similar results figures together. For example, the same laser time and laser power of cancer and non-tumor cells data should be in one figure (Figure 2a and Figure 2b…).
- The concentration-dependent of GNR and silica nanoparticles treated cells should be added in the manuscript.
- The cell mortality rate is different between tumor and non-tumor cells, and the authors did not use any targeting ligands for better internalization. Why is in-vitro cell viability different in the case of localized administration of GNR and silica nanoparticles?
- Supplementary Materials: The following are available Figure S1: 484 titles, Table S1: title, Video S1. However, the review did not find any Supplementary Materials for review. The authors can add the SEM or TEM images for GNR and silica nanoparticles and its size distribution.
Author Response
We have generated a document for the reviewer

Round 2
Reviewer 1 Report
The authors have addressed all relevant comments adequately. In my opinion, it can be accepted for publication.
Reviewer 2 Report
The current version is suitable for acceptance.